# Strategies to Support Sustained Participant Engagement in an Oral Health Promotion Study for Indigenous Children and Their Families in Australia

**DOI:** 10.3390/ijerph19138112

**Published:** 2022-07-01

**Authors:** Megan L. Hammersley, Joanne Hedges, Brianna F. Poirier, Lisa M. Jamieson, Lisa G. Smithers

**Affiliations:** 1School of Health & Society, Faculty of the Arts, Social Sciences and Humanities, University of Wollongong, Wollongong, NSW 2522, Australia; mhammers@uow.edu.au; 2Early Start, Faculty of the Arts, Social Sciences and Humanities, University of Wollongong, Wollongong, NSW 2522, Australia; 3Illawarra Health and Medical Research Institute, Wollongong, NSW 2500, Australia; 4Australian Research Centre for Population Oral Health, Adelaide Dental School, University of Adelaide, Adelaide, SA 5005, Australia; joanne.hedges@adelaide.edu.au (J.H.); brianna.poirier@adelaide.edu.au (B.F.P.); lisa.jamieson@adelaide.edu.au (L.M.J.); 5School of Public Health, University of Adelaide, Adelaide, SA 5005, Australia

**Keywords:** Indigenous, Aboriginal, engagement, research, retention

## Abstract

The health inequities of Indigenous peoples compared with non-Indigenous peoples are significant and long-standing across many countries. Colonisation and dispossession of land and culture has led to profound and devastating consequences on the health of Indigenous peoples. A lack of trust and cultural security of health services remains a barrier to participation in health care services. Similarly, engagement in research activities is also hindered by a history of unethical research practices. Creating partnerships between researchers and Indigenous communities is key in developing research studies that are culturally appropriate, acceptable and relevant to the needs of Indigenous peoples. Baby Teeth Talk was a randomised controlled trial conducted with Indigenous children and their mothers in South Australia that tested an intervention involving dental care, anticipatory guidance on oral health and dietary intake, and motivational interviewing. The study was developed in consultation and partnership with local Indigenous communities in South Australia and overseen by the study’s Aboriginal reference group. The recruitment and retention of participants in the study has been strong over numerous waves of follow-up. The purpose of this paper is to describe the strategies employed in the study that contributed to the successful and sustained engagement of the participants. These strategies included the establishment of an Aboriginal reference group, building relationships with organisations and community, flexibility of appointment scheduling and allocating adequate time, reimbursement for participant time, developing rapport with participants, encouraging participant self-determination, and adaptation of dietary data collection to better suit participants.

## 1. Introduction

The health gap between Indigenous (Indigenous is defined as *“Tribal peoples in independent countries whose social, cultural, and economic conditions distinguish them from other section of the national community and whose status is regulated wholly or partly by their own customs or traditions or by special laws or regulations; and peoples in independent countries who are regarded as Indigenous because of their descent from the populations who inhabited the country, or a geographical region to which the country belongs, at the time of conquest or colonisation or the establishment of present state boundaries and who, irrespective of their legal status, retain some or all of their own social, economic, cultural, and political institutions”* [1]) and non-Indigenous peoples worldwide is well documented. Such health inequities stem from a history of colonisation and dispossession of land and culture across many generations [2]. However, despite these impacts, Indigenous peoples and cultures have endured, resisting forced assimilation, with Aboriginal and Torres Strait Islander peoples in Australia (respectfully referred to as ‘Indigenous’ hereon in this paper) being among the longest continuing cultures in the world [3]. Oral health is of particular concern, with a significantly higher level of dental caries experienced by Indigenous populations [4]. In Australia, untreated carious lesions were present in 44% of Indigenous children aged 5–10 years in 2012–2014, compared with 26% among non-Indigenous children [5]. Furthermore, intake of energy from added sugar, which is a major contributor to dental caries, is higher in Indigenous children than non-Indigenous children [6]. Although there are complex social and political factors at play that impact on education and employment opportunities, adequate housing, and access to health services, this inequality is also partly attributed to the inability of existing oral health and other health services to meet the cultural needs of Indigenous communities [7]. Barriers to participation in health care services also stem from the profound and intergenerational effects of colonisation, which has resulted in many Indigenous peoples having an entrenched mistrust, from having endured poor treatment and racism over many years [8,9]. It has been recognised that more culturally appropriate child oral health promotion programs are needed [10]. Efficacy testing of programs for Indigenous populations is often hindered by generally low engagement and retention rates, given the commonality of past unethical and exploitive research practices with Indigenous communities. Past research has often been conducted by non-Indigenous researchers for their own interests, rather than being led by Indigenous peoples and grounded in community needs, where there are benefits from shared knowledge and Indigenous peoples learning about research [9].

Baby Teeth Talk was an oral health promotion randomised controlled trial for mothers of Indigenous children. It was developed in partnership with local South Australian Indigenous communities and an Aboriginal Reference Group that was formed specifically for the study. Briefly, mothers pregnant with an Indigenous child or having recently given birth (baby less than 6 weeks of age) who were residents of South Australia (*n* = 448) were recruited to the study during 2011–2012, and provided written informed consent. Participants were recruited from several sources such as Indigenous groups, community services, and hospitals. Posters were placed in maternity hospitals and Aboriginal Community Controlled Health Services. Stakeholder engagement in maternity hospitals assisted in spreading the word about the study and referral forms were used to allow the research team to follow-up on potential participants. Snowballing was also used as a recruitment strategy. The intervention consisted of four components comprising maternal oral health care for mothers during their pregnancy, child fluoride varnish application, anticipatory guidance on oral health and dietary intake, and motivational interviewing, that resulted in lower carious lesions at 2, 3 and 5 years of age in the intervention group compared with control. More details of the methods and results of this study have been described elsewhere [11,12,13]. The proportion of eligible mothers recruited to the study was approximately two-thirds of the target population, and an exceptional retention rate of 74% at the 2-year follow-up was achieved (please refer to Figure 1 below), substantially higher than some other oral health studies in Indigenous populations in Australia [14,15]. While the retention rate is slightly less than in another Australian study [16], and a study in North America [17], it should be noted that the participants in the Baby Teeth Talk study were generally living in more geographically dispersed regions and not necessarily in specific Aboriginal communities, as was the case in other studies. For example, this study entailed travelling 700 km west, 400 km east and 800 km north. This geographical spread therefore imposed significant challenges in regard to the recruitment and retention of participants. The study is ongoing, with children of mothers enrolled during pregnancy now aged 10–11 years.

The aim of this paper is to explore and describe the strategies employed in the Baby Teeth Talk study that contributed to the successful and sustained engagement of the participants, which supported the child and carer in improving their oral health outcomes and knowledge. Members of the research team provide detailed reflections on the elements of the study that they perceive contributed to the high level of recruitment and continued involvement of participants over time.

## 2. Perceived Factors Contributing to Successful Recruitment and Sustained Engagement of Participants

### 2.1. Development of the Aboriginal Reference Group

The Aboriginal Reference Group established for this study was critical to the overall success of the project. The Aboriginal Reference Group comprised Indigenous leaders in several areas of health, such as the Local Health Networks of South Australia and Aboriginal Community Controlled Health Services. The group initially comprised the Chair of the existing Indigenous Reference Group from the Indigenous Oral Health Unit at the University of Adelaide. When developing Indigenous research, Indigenous governance is an ethical obligation to ensure that non-Indigenous researchers working with Indigenous populations act in a culturally secure way. Hence, the lead researcher of the project engaged and invited key Indigenous leaders within the health sector to provide leadership and governance of the project. The Aboriginal Reference Group provided oversight and guidance of research design, development of measurement tools, intervention content, implementation, and evaluation. The group also piloted and provided feedback on all questionnaires used. Furthermore, they provided feedback on the research plan from an Indigenous perspective, ensured it was culturally appropriate, secure, and determined safe ways of conducting research with communities while considering research outcomes and future translation of findings. When discussing the project with mothers at recruitment, the presence of an Aboriginal Reference Group overseeing the study provided credibility and facilitated the formation of trust with the research team.

### 2.2. Building Relationships with Health Organisations and Local Communities

Establishing and strengthening relationships between the research team, oral health care services, Aboriginal Community Controlled Health Services and other local health organisations, as well as community members, was crucial to ensure wide engagement and dissemination of information about the study to Indigenous mothers. The team delivered presentations to the Women’s and Children’s Hospital in Adelaide, hospitals in greater Adelaide and the Maternal Child Health Nurse Network, and spoke with and distributed study information to nurses, midwives, Aboriginal health workers, and Aboriginal maternity workers. In particular, the relationships established with Aboriginal Community Controlled Health Services and Aboriginal health workers were critical to successful community engagement. There was a wealth of support from the various stakeholders as the goals of the study were clearly communicated and stakeholders highly valued the importance of improving child oral health. Community networks were also utilised, with word of mouth being a major avenue for recruitment. Other communication opportunities in communities were explored to ensure that as many community members knew about the study as possible, including women’s groups, art groups, cultural groups, and men’s groups (who may have had partners who were pregnant). In all communications about the project, the individual and community level benefits of participation in the project were emphasised, which was essential for stakeholder and participant buy-in. Such benefits included provision of dental care to mothers during pregnancy, with dental care subsequently offered to everyone in the family, tailored guidance for mothers in relation to their child’s oral health, and sharing study progress and findings with the community. Another benefit was the partnership established with SA Dental (state government dental service), which ensured that children were having regular dental visits and gained familiarity with the dental service environment in their local area.

Communication and connections with community members and organisations were central to participant retention in this study. The information provided through these connections regarding participant whereabouts, ‘sorry business’ (i.e., cultural practices following the death of a family member), and family business, among other insights, proved to be invaluable both for locating participants as well as respecting participant needs and culture. This information helped to determine if it was appropriate to delay visits until another time.

During study visits, family were asked if oral health care was needed by any family member, providing a conduit to services; this practice also extended the benefits of participation beyond mother–child dyads. Many Indigenous peoples and women of childbearing age can find it challenging accessing oral health care services. The study team continually reinforced to participants that the best possible care was wanted for the participants and their families, and they were encouraged to contact the study team at any point in the future if they had issues accessing services. It is perceived that facilitation of oral health care appointments as well as accessing these services from a financial perspective is one of the factors that motivated mothers to enrol and remain in the study.

Reciprocity was considered imperative in retaining participants in the study. Often the lag between data collection and publication of results is considerable, but indicators such as the number of people seen at different time points and some brief results were fed back to the community through newsletters and presentations to Aboriginal Community Controlled Health Services. However, it is acknowledged that improvements could be made, and future studies should strive to do this in a more timely manner.

### 2.3. Providing Reimbursement for Time and Expertise

Providing reimbursement for time and expertise was an important part of the study to recognise the generous and voluntary participant contributions to the project, and also to provide participants with the resources to implement the anticipatory advice provided in the sessions. Toothbrushes and toothpaste were provided to the family and an AUD 50 gift card for use at a major supermarket chain across South Australia was offered at each visit. This provided families with options to purchase varied fresh and healthy packaged foods and while it wasn’t specified what this money was spent on, it is thought that this may have assisted families to make healthy eating purchases in line with the anticipatory guidance provided.

### 2.4. Flexibility of Appointment Scheduling and Allocating Adequate Time

The majority of the Baby Teeth Talk intervention and follow-up appointments were conducted in participants’ homes. Conducting the study in centralised locations such as oral health care services or at a University would very likely have been a barrier to participation. Many of the participants lived in rural and remote areas of South Australia, requiring the research team to undertake lengthy regional trips. The visits themselves were up to 2–3 h duration, and additionally there were often many hours of phone calls, leaving messages, sending text messages, and at times making visits to houses trying to make contact and schedule appointments. It was often very difficult to find people, with participants changing address and phone number. Another common issue encountered was participants having no phone credit, so they had no means to call or text the study team back. At baseline and follow-up appointments, names and phone numbers of alternate contacts were collected that were sometimes very effective in making contact. Some participants had moved during the study and at times the team visited even more remote locations outside of the state of South Australia to facilitate ongoing participation. There was a need to be flexible and understanding that participants have other priorities in their lives. Appointments were not made too far in advance, usually after coming into a town. With appointments made within this short timeframe, it is thought that participants were more likely to be available at the time of the appointment.

It is acknowledged that, often, similar studies do not have sufficient resources for the type of time commitment afforded in this study to recruit, follow up and spend time with participants, and that the rigour of clinical trials often does not allow provision for a great deal of flexibility. However, running to a strict time schedule is not conducive to establishing relationships with Indigenous peoples. Indigenous peoples want to be able to share information and feel that they are being listened to. Having the time to dedicate to this will result in the Indigenous person leaving the appointment with health information that they have processed, making them feel that they are more in control. In order to be conducted optimally, research funding for projects involving Indigenous participants should consider these factors and allow for additional time and resources, and flexibility of implementation timeframes [18].

### 2.5. Developing Rapport and Trust with Participants

Several strategies were employed to build and maintain relationships with participants, aligning with the core values of the National Health and Medical Research Council Ethical Conduct with Aboriginal and Torres Strait Islander Peoples and Communities Guidelines of spirit and integrity, reciprocity, respect, equality, survival and protection, and responsibility [19]. Cultural safety was a priority, ensuring that the intervention provided was safe and culturally appropriate (encompassing a holistic view of health that considers community, spiritual, emotional, physical, and environmental factors), and that shared respect, knowledge and meaning was demonstrated during all communication and interactions [20,21,22]. One of the major strengths of this study is the respectful attitude of, and commitment to cultural safety/security by, the research team. Cultural competency training was completed by all researchers prior to project commencement and refresher training was provided in subsequent years. Indigenous peoples participating in the research were comfortable to continue with the study as they felt safe in the way they were respected when researchers visited their home. Supporting cultural safety throughout the study assisted in building trust and rapport, and made participants feel valued and respected and that their individual needs and the needs of their community were of upmost importance to the research team.

Data collection was designed to be engaging, rather than straight question and answers on paper. For example, a food model booklet was used to enable participants to have a visual to refer to when answering the dietary intake questions. Locally acquired child drinking cups were also used to more accurately estimate fluid intake. Use of these visual resources allowed for inclusion of all participants. Another benefit of the use of more visual and storytelling knowledge translation techniques was their cultural appropriateness. Furthermore, specific needs and issues that were identified by participants throughout the study were considered carefully and acted upon. For example, it was intended to complete 24 h food recalls at the 2-year and 3-year data collection timepoints, but feedback obtained from participants indicated that the burden of these recalls was substantial and as a result, the dietary intake data collection method was adapted to a food frequency questionnaire from the 3-year timepoint onwards. It is important that obtaining optimal data is balanced with potential disengagement and withdrawal of participants.

Members of the research team documented details on important events and situations happening within the family at the time of the visits, for example the recent birth of a baby or a family member being unwell. These details were reviewed before each visit to ensure that the researcher attending the home was well informed about any issues and was prepared about what to expect and helped to develop a sense of caring through the researcher asking participants about recent events or illnesses. Being aware of these details was critical for new research staff joining the team. The research team also maintained communication with the family in between visits, which included sending a personal note to the family to thank them for having the researcher in their home and sending a birthday message to mothers participating in the project.

Although not adopted as a specific methodology, Relational Yarning was used, which was integral in establishing and maintaining relationships with participants throughout the duration of the study. For Indigenous peoples in Australia “Yarning” refers to conversation, storytelling, or the sharing of knowledge. Relational Yarning refers to *“Yarns that happen alongside research as a by-product of spending time with community members for research purposes. These yarns are not central to the research project at hand, but we view them as necessary to meaningful and ethical Indigenous research processes”* [18]. Poirier et al. [18] outlined six core values that are central to Relational Yarning; respect, relationships, advocacy, reciprocity, time, and gratitude. Relational Yarning helps to achieve and maintain these values and prioritise them in research interactions. In the Baby Teeth Talk study, the Relational Yarning values were integral in establishing relationships. The approach taken to asking participants the questions was very personable and relaxed and would often result in the discussion moving to tangential topics of interest to participants. Participants directed conversations around general prompts and were respected and acknowledged as the experts of their own lives. Conversations worked to empower participants, in alignment with the motivational interviewing methodologies employed, and worked to eliminate power imbalances all too common in Indigenous health research. It was important that individuals felt that they were valued and listened to. Family was a very notable focus of discussion for many participants; as well as their own family, they wanted to learn about the family backgrounds of the researchers. The research team entered the homes of the participants with a very respectful and open mind, and the researchers perceived that participants felt that they were listened to and therefore were comfortable to invite the researchers back into their homes at later timepoints. The breadth of experience of key team members with Indigenous communities was a critical factor in establishing trust with other researchers, but essential to all was a sense that it was a privilege to be involved in the study and that Indigenous participants/communities were the knowledge holders (not the researchers/research assistants).

Relational Yarning facilitates relationship building and prioritising respect for individuals and the community above research metrics. It is important to note that achieving study outcomes are not the only indicators of success. Enabling engagement and developing trust between researchers and communities is critical to establishing solid foundations for future research relationships, but such efforts are largely unrecognised by academic metrics such as paper publications and citations. Relational Yarning was complemented by the Motivational Interviewing method used in this study. Motivational Interviewing is a method of counselling that is underpinned by a collaborative relationship between the interviewer and interviewee [23]. Motivational Interviewing is a powerful means of communicating with people and is compatible with establishing rapport and trust. The core emphasis is on reflective listening with empathy, where the interviewer acts as the facilitator, helping to empower the interviewee, rather than telling them what they should do. People who are ambivalent about a particular health behaviour are assisted to identify discrepancies between the current status quo and where they would like to be, while providing them with autonomy and developing their own intrinsic motivation to change. Originating in the substance addiction field [24], Motivational Interviewing has been used successfully in several areas of health [23], including oral health [25]. The Baby Teeth Talk study was the first study where Motivational Interviewing has been used in Australia with Indigenous communities in an oral health context. Specifically, it was used over four sessions from pregnancy up to child age of 18 months to: encourage mothers to attend oral health care appointments during pregnancy, promote the importance of offering children non-cariogenic foods and drinks, highlight the importance of fluoride application, and encourage mothers to take their child for an oral health check [12]. Motivational interviewing and anticipatory guidance were offered to parents in an individualised way that took into account specific concerns, barriers, and support structures. There was a motivational interviewing guide that included general topics to be covered specific to each time point. The intent of Motivational Interviewing is to garner participant interest in changing behaviour and sue participant identified motivations to encourage behaviour change. Further details on the motivational interviewing approach used in this study can be found in previously published papers [11,12].

### 2.6. Encouraging Participant Self-Determination

Efforts were made throughout the study to support self-determination. Self-determination refers to the ability to make one’s own decisions, which can influence the amount of control they feel they have with their life, and autonomous motivation to make changes rather than being felt pressured to make changes from external influences [26]. The self-determination theory also complements Motivational Interviewing [27]. An example of where self-determination was supported throughout the study involved encouraging the mothers to hold food packaging when discussing how to read food labels, with the intent to improve the mother’s self-efficacy in determining whether the product contained excessive sugar. Self-determination was also supported by encouraging participants to complete their own questionnaire where possible. This was important to give participants a sense of ownership and control over the information that they were providing. If the participants were not able to complete the questionnaire, questions were re-worded, or family members assisted with translation. Participants reflected at the end of each session on what they had gained from their discussions and developed an oral health goal for their child based on their new knowledge and personal priorities. Asking participants to reflect on the discussion helped to reinforce the information and also to consider if they were going to choose to do something differently with that new knowledge. There were a range of strategies used that aimed to increase participant knowledge at each follow-up, including hands-on nutrition label reading of baby foods and drinks, how to use a toothbrush, and other practical oral health skills that the carer were interested in learning about. Visual resources were also used, and oral health care worksheets were completed that carers kept with them as a reminder of their behaviour change plan. Examples of health goals included making healthier food choices, through reducing sugar sweetened beverage consumption or selecting products with less sugar content, which was facilitated by the participant’s increased knowledge of label reading. Participants were also asked for permission to return for the next visit, giving control and ownership to the participant, and emphasising that the interaction was not one-sided.

## 3. Discussion

Indigenous children’s diet and oral health are critical to child well-being, as such, concerted efforts are needed to overcome barriers experienced by Indigenous communities to consuming healthy diets and accessing appropriate oral health care. Creating partnerships with Indigenous communities is critical to developing nutrition and oral health research and improved provision of services that are culturally appropriate and relevant to the needs of the communities.

There is also potential for this research to impact participants in a broader sense. There are many anecdotes from the members of the research team that demonstrate the improvement in self-efficacy and empowerment among the participants in regard to oral health prevention practices. It is also recognised that such empowerment has the potential to cross over into other areas of health care. It is therefore anticipated that participants may have more confidence to raise concerns if they have faced negative experiences in a health care setting based on the increased self-efficacy that they have achieved through the intervention.

While many positive aspects of the study have been identified in this research, there are of course barriers that were encountered. Despite considerable effort and persistence, there were some participants who could not be contacted for follow-up due to relocation and change of phone number. Future studies could be aided by collecting additional alternative contact details at baseline. It has also been identified by our research team that Aboriginal health workers from Community Controlled Health Services, Aboriginal community organisations, community leaders and community members hold valuable knowledge about oral health and the community. Engaging Aboriginal health workers to become part of the research team collecting data could be a means to more easily locate participants as Aboriginal health workers are regularly engaged with local communities. This would also provide an opportunity for two-way sharing of knowledge and provide Aboriginal health workers with upskilling in oral health care. Many of the strategies that have been identified have now been adopted by the research group and now form part of how research is conducted. The Baby Teeth Talk research was overseen by the Aboriginal Reference Group, who were critical in guiding the research design, development of measurement tools, intervention content, implementation, and evaluation. Future studies may benefit from a deeper level of engagement with the community, such as through community-based participatory research to form sustained partnerships with embedded Indigenous leadership in the project team, engaging the community from the point of study inception and design.

## 4. Conclusions

This paper outlined several strategies adopted in the Baby Teeth Talk study that the authors consider integral to recruiting and retaining Indigenous research participants, but most importantly are integral to encouraging healthy diets and maintaining oral health for Indigenous peoples across one’s life course. It is recommended that future studies seek to engage Aboriginal Community Controlled Health Services and community members as project partners, as well as establish an Indigenous Reference Group. It is also imperative that sufficient time and resources be allocated to ensure meaningful engagement with participants, allowing for optimal time to devote to participant interviews and communication, essential for establishing rapport and trust. Consideration could also be given to using Motivational Interviewing and Relational Yarning, which were complementary tools in supporting ongoing participant engagement throughout this study. The foundational values and strategies employed in the Baby Teeth Talk project could extend beyond Indigenous dietary and oral health research to other areas of inquiry related to Indigenous wellbeing.

## Figures and Tables

**Figure 1 ijerph-19-08112-f001:**
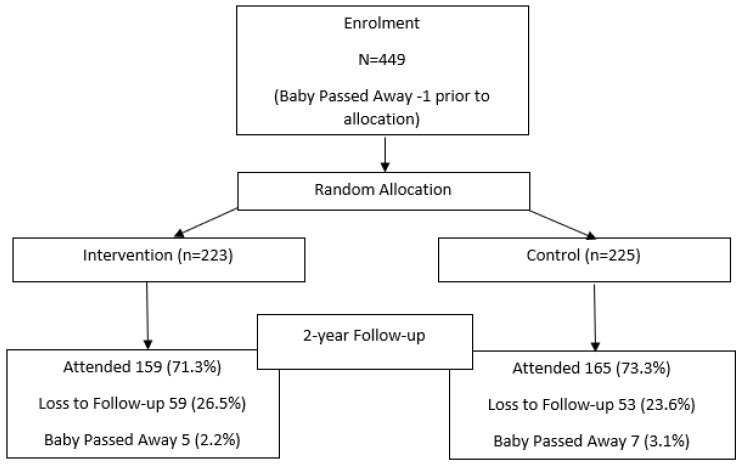
Baby Teeth Talk study participant recruitment, allocation and 2-year retention.

## Data Availability

The data presented in this study are available upon reasonable request from the corresponding author. The data are not publicly available due to conditions of ethics approval.

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
