# Peer review of "Strategies to Support Sustained Participant Engagement in an Oral Health Promotion Study for Indigenous Children and Their Families in Australia"

_ijerph, 2022, doi:10.3390/ijerph19138112_

Round 1
Reviewer 1 Report
The authors raised an important topic. Insufficient access to health care, lack of awareness of the need for dental treatment and prophylaxis are the reasons for the development of oral cavity diseases
1.Line 54 - What complex factors do the authors mean?
2.line 62- what unethical research practices with Indigenous communities ?
3. The aim of the study is mentioned only in the abstract. The authors may add a separate paragraph after the introduction.
4. line 85 How was the group for the study created? Who was in the group? What were the recruitment rules? Were there including and excluding criteria?
5. line 110 -“with word of mouth being a major avenue for recruitment “Were there any other methods of reaching the study group?
6. line 114 . what benefits?
7. How many of the initially involved in the study remain or are still under control?
8. Did the participants sign their consent to participate in the study?
9. The authors emphasize that an individual approach is important, but did they use any forms or questionnaires for research? How was participant involvement assessed and compared?
10. Where participants' data were stored, information about their lives, who had access to them and how they were secured?
11. line 232 “researchers perceived that participants felt that they were listened”- how did they perceived? Was it just their subjective opinion?
12. What was the motivational interview like, was there a scheme? Did the authors check the level of knowledge and the participants' interest in improving their oral health?
13.line 278 How was knowledge increased? eg word of mouth,brochures, books?
14. The authors should add a paragraph- Discussion
15. 288- rather for discussion
16. “The purpose of this paper is to describe the strategies employed in the study that contributed to the successful and sustained engagement of the participants”- The conclusions do not relate to the purpose of the paper.
Author Response
Thank you for taking the time to review our manuscript. Please find our response to your feedback attached.

Reviewer 2 Report
Dear Authors,
I have read your article with some interest. I must admit it is not exactly my area of expertise as the topic is focused on improving retention of aboriginal participants in a randomised control trial with oral health interventions.
I have a few comments. Firstly, you comment about your retention rate and say it is higher than comparable studies. Can you provide more precise numbers, and possibly display these key metrics graphically. This is the thrust of the paper and the reader has to be reading very attentively to actually gather this information in its present form. Furthermore, this study by Slade et al. https://onlinelibrary.wiley.com/doi/full/10.1111/j.1600-0528.2010.00561.x showed only about 20% loss to follow up at 2 years, which is better than the present value of 28% (72% retention). How do we know these methods are actually helpful if similar studies get better retention without the increased time and financial burden associated with the methods described here?
Second - many of the interventions such as diet diaries and reading food product labels require reading. My understanding is that literacy is extremely poor in this population with maybe half or more of adults being illiterate.
see: https://link.springer.com/article/10.1007/s13384-020-00388-7#Sec8
This seems like something you should make reference to and explore in your work. It would seem to me to be a major barrier to recruitment yet you don't seem to make any mention of this.
Lastly, you talk about the steps you have made to ensure your work is "culturally appropriate". What does this actually mean? Can you give some examples of appropriate/inappropriate aspects relevant to this culture?
Author Response

(The authors gave the same response as above.)

Reviewer 3 Report
It is a well-written paper on a very interesting topic in terms of strategies to support sustained participant engagement in an oral health promotion study in Australia. The authors shared valuable experience and strategies to achieve a high retention rate of participants in a long-term study. I believe it would be helpful for future researchers to conduct studies within the community.
The concept to make community engaged in academic research become more and more popular, which is often named as community-based participatory research. Academia do not play a dominant role any longer, while different parties, such as the community, the stakeholders and the academic, work in collaborations to identify a research gap, design and implement the study. It would be informative that the authors bring up this concept in the paper.
It is no doubt that successful strategies should be shared and passed to future researchers, but barriers and failures are worthy to be discussed as well. It would be interesting to discuss and explore difficulties and barrier in implementing this community engaged study, and how they were dealt with. Recommendations can be given to future researcher to make their project successful.
Author Response

(The authors gave the same response as above.)

Round 2
Reviewer 1 Report
The authors responded to the comments in the review and sufficiently corrected the manuscript.Thank you for answering my questions.
Reviewer 2 Report
the concerns from the first review have been satisfied